# Enhancing student career guidance and sentimental analysis: A performance-driven hybrid learning approach with feature ranking

Sana Yaqoob[1], Ayman Noor[2], Talal H. Noor[2], Mohammad Zubair Khan[3]*, Anmol Ejaz[4], Md Imran Alam[5], Nadim Rana[6], Khurram Ejaz[1]*

1 Department of Computer Science and Information Technology, University of Lahore, Lahore, Pakistan, 2 Department of Computer Science, College of Computer Science and Engineering, Taibah University, Madinah, Saudi Arabia, 3 Department of Computer Science and Information, Applied College, Taibah University, Madinah, Saudi Arabia, 4 Department of Healthcare Administration and Management at the University of the Potomac in Virginia, Virginia, Virginia, United States of America, 5 Department of Electrical and Electronics Engineering, College of Engineering and Computer Science, Jazan University, Jazan, Saudi Arabia, 6 Department of Computer Science, College of Engineering and Computer Science, Jazan University, Jazan, Saudi Arabia

* khurram.ejaz@cs.uol.edu.pk (KE), mkhanb@taibahu.edu.sa (MZK)

## Abstract

Choosing the appropriate career path poses a significant hurdle for students, especially when time is constrained. This research addresses the challenge of career prediction by introducing a method that integrates additional attributes, refines feature prioritization, and streamlines feature selection to enhance prediction precision. The key objectives of this study are to pinpoint pertinent features, accurately rank them, and enhance prediction accuracy by eliminating non-essential features. To accomplish these aims, three methodologies are employed: Feature Fusion and Normalization (FFN) for precise data identification, Average Feature Ranking (AFR) utilizing a blend of Random Forest (RF) and Linear Regression (LR) for feature prioritization, and Improved Prediction with Weighted Characteristics (PWF) which integrates Principal Component (PC) analysis for feature reduction. The prediction performance is assessed using a hybrid Multilayer Perceptron (MLP) classifier with 5-fold cross-validation. The outcomes reveal that the hybrid approach yields a superior feature set for prediction. The top twelve ranked features are determined by averaging each feature's RF scores and coefficients. The achieved accuracy (ACC), precision (P), recall (R), and F1 scores stand at 87%, 87%, 86%, and 86%, respectively, with an Area Under the Receiver Operating Characteristic Curve (AUC-ROC) value of 92%. These findings underscore the efficacy of the proposed hybrid learning technique in accurately forecasting career trajectories.

**Data availability statement:** All relevant data are within the paper.

**Funding:** The author(s) received no specific funding for this work.

**Competing interests:** The authors have declared that no competing interests exist.

## 1. Introduction

Forecasting student achievement is growing in modern education, especially in developing nations, as it directly influences the educational framework. Recognizing students who might encounter difficulties and earn low grades enables educational establishments to take preemptive actions, like providing extra lectures or sessions to aid their academic advancement. Utilizing existing information and data, educational institutions can adeptly cater to the requirements of individual students, thus elevating the overall standard of education [1].

A student's skills encompass their knowledge and abilities, and these features interest the psychology department at the University of Lahore. These skills can be classified into four skills: cognitive skills, learning style, comfort level, and problem-solving skills. Cognitive skills refer to an individual's diverse methods of perceiving, thinking, and remembering information. Learning style encompasses various components and can be categorized into different types, such as the yes/no variety, student behaviors, and attitudes. Comfort level relates to students' ease or familiarity with their course materials. Problem-solving skills indicate an individual's proficiency in listening, reading, writing, and decision-making [2].

In the realm of feature learning, accurately pinpointing data containing relevant features that enhance model performance poses a significant challenge. Numerous investigations have delved into dimensional reduction techniques, such as feature selection and extraction methods, aiming to enhance prediction and model efficacy [3,4]. This research proposes a novel approach that enriches a dataset with new features and preprocesses the data through categorization, labeling, and normalization of features. Additionally, Random Forest (RF) and Linear Regression (LR) are employed to identify the most influential features. To further refine prediction accuracy, Principal Component (PC) analysis is utilized to isolate weighted features that contribute to performance enhancement. Classification tasks are undertaken using Multilayer Perceptron (MLP), while the effectiveness of predictions is assessed through 5-fold cross-validation.

Integrating a new feature necessitates updating the dataset, specifically customized for sentiment analysis. By utilizing a feature reduction technique employing a feature ranking method, we attain precise classification with an 87 percent accuracy rate. This enhancement aligns with the adjustment of coefficients linked to each sentimental feature. The third advancement introduces a hybrid method, leveraging the ReLU activation of the multi-layer perceptron as a crucial catalyst. This activation function additionally trims features, leading to an outstanding accuracy of around 95 percent.

The remaining sections of the paper are structured as follows: Section II provides an overview of the literature. Section III delves into the research methodology. Section IV discusses the results obtained, V gives an illustration of comparative analysis, and Section VI serves as the conclusion, summarizing the work and pointing toward future directions.

## 2. Literature review

Two recommendation systems, "CareerRec" [5] and "ACCBOX" [6], forecast the career trajectories of IT students. CareerRec, leveraging a dataset of 2255 IT

students, achieves a 70% accuracy employing XGBoost, outshining KNN, DT, Bagging, and GB. Conversely, ACCBOX, introduced by Kehindethe et al., attains a 63% accuracy, outpacing SVM, DT, RF, Logistic Regression, and XGBoost, relying on student skills and economic status. Alsalman and Tree [7] proposed a model to bolster students' skills grounded on past academic performance, utilizing data from five engineering streams. The model, evaluated with KNN, SVM, and ANN, showcases remarkable performance, particularly with an impressive 97% accuracy attained by the Artificial Neural Network (ANN).

Zaffar et al. [8] employed an ANN to forecast students' CGPA, achieving 95% and 85% accuracy on two test sets. Another study introduced an ML model for GPA prediction; ANN attained the highest accuracy at 97%, while DT scored lower at 66%, highlighting the importance of course failure time [9]. Rimadana et al. [10] applied C4.5 algorithm to predict employment trends, reaching an 81% accuracy rate with 420 samples. Nazareno et al. [11] investigated optimal feature selection with various algorithms across different datasets. Thirteen classification techniques were tested, with MLP and SMO yielding superior recall, precision, and F-measure results. Their findings showcased a significant 10% to 20% accuracy improvement through diverse datasets and feature selection methods.

Ahmad and Shahzadi [12] developed a student performance prediction system utilizing the Levenberg-Marquardt algorithm within a neural network framework, incorporating demographic factors and academic records. Their model achieved an accuracy of 84%, with specificity at 54%, sensitivity at 94%, and precision of 86%. Evaluation via the ROC curve indicated an AUC value of 86%. Yadav et al. [13] utilized machine learning models to predict student performance based on time management skills, employing a dataset of 125 students with 11 questionnaire items. Neural networks achieved 75% accuracy in academic scores and 81% in English performance, while decision trees yielded 68% accuracy in academics and 71% in English. In another study in [14], a CGPA forecasting model was proposed using socioeconomic background data from undergraduates, achieving an 84% accuracy with artificial neural networks and an AUC value of 86%.

In a related study in [15], an MLP classification model with five features achieved 95% training accuracy and 75% holdout accuracy in distinguishing 'at risk' and 'not at risk' student degrees. Aydoğdu's CorC-Net Neural Network model outperformed other algorithms with 92% accuracy via cross-validation. Delen et al. [16] proposed a student performance prediction model using the chi-square method for feature reduction, selecting high-value features through open questions. Among five ML classifiers employed, ANN reached 92% accuracy, while KNN reached 37% after feature reduction. Ghorbani et al. [17,18] introduced an MLP-based model for student profile skill set calculation, achieving 62% accuracy with MLP and 58% with DT after feature reduction using Information Gain (entropy).

Kukkar et al. [19] developed an Artificial Neural Network (ANN) model predicting student performance in e-learning, achieving 80% accuracy in the fall semester of 2017–2018. Key contributors to success included attendance, course attendance, and time spent. Python as one of the prominent languages in research [20,21] with TensorFlow and Keras [22] aiding precise predictions. Gelbard et al. [23] employed ontology criteria for personality determination, and an attention-based CNN and Deep learning model-based sentiment identification. Similarly, Basiri et al. [24] introduced a novel deep-learning architecture to enhance sentiment analysis tasks. The proposed model, ABCDM, integrates bidirectional Long Short-Term Memory (LSTM) and Gated Recurrent Unit (GRU) layers to capture past and future contextual information within textual data. To address the challenge of high-dimensional feature spaces and the uniform weighting of features in traditional models, the authors incorporate an attention mechanism that assigns varying importance to different words, thereby refining feature selection. Convolution and pooling layers reduce feature dimensionality and extract position-invariant local features. Evaluated across five review datasets and three Twitter datasets, ABCDM demonstrated superior performance compared to six recently proposed deep neural networks, achieving state-of-the-art results in both long-form review and short-form tweet sentiment classification.

D. Haritha [25] proposed a machine learning-based system to help students in the CSE/IT domain choose suitable career paths. The system uses algorithms like Decision Trees, Neural Networks, and Collaborative Filtering to provide personalized course recommendations based on student skills and interests. While the approach highlights the importance

of aligning education with individual aptitude, the paper lacks detailed evaluation results and broader applicability beyond technical fields. Andrade et al. [26] advocated for diverse features in career profiling. A decision tree-based technique for predicting employment patterns achieves 81% accuracy [27] and the C4.5 algorithm classification yields 81% accuracy [28]. An instinctive career system, using KNN (73%) and Stochastic Gradient Descent (81%), matches students' answers to aptitudes and personalities [29].

Prakash and colleagues [30] developed a model using a comprehensive dataset to boost model accuracy through an examination of the AUC-ROC metric, employing K-fold cross-validation to mitigate overfitting. They utilized a dataset divided into five subsets for testing and training, implementing the CatBoost model alongside fivefold cross-validation. The AUC-ROC values, ranging from 0.68 to 0.85, indicated commendable performance. In a separate study centered on psychological aspects of student career guidance, an 8-puzzle game and aptitude/IQ tests were utilized [31]. Zahour et al. [32] collected data via a 36-question questionnaire for their CPRES model, while Lauet et al. [33] introduced a framework for evaluating students' abilities through tests, with results transformed into numerical scores. Employing a machine learning approach with Artificial Neural Networks (ANN), they achieved a 74% accuracy rate in forecasting outcomes based on academic performance, talent, employment history, and personality traits [34].

Atienza, John Robert D., et al. [35] introduced an artificial neural network (ANN) system designed to forecast student performance by leveraging demographic information and previous academic achievements, achieving an accuracy rate of 84%. Mengash [36], on the other hand, delved into career counseling, pinpointing personality traits crucial for precise career prediction, with the "thinking/feeling" aspect exhibiting the highest accuracy at 92%. Sandoval-palis et al. [37] employed an ANN to anticipate student outcomes, yielding a 92% accuracy rate in candidate selection. The results show that the Area Under the Curve (AUC) metric consistently served as a benchmark for assessing the reliability of ANN predictions and classifications. ANNs have proven adept at assessing students' potential in computer science [38] one approach integrates academic, personality, and personal data (86% AUC), while another utilizes backpropagation training (94% accuracy) [39]. Regarding predicting success, an ANN model based on high school grades, entrance exams, and aptitude scores demonstrates well-rounded performance (79% accuracy, 78% recall, 81% precision, 79% F1 measure) [40].

Okubo et al. [41] proposed an artificial neural network (ANN) model to forecast student performance, trained with variables such as score, Liability Index, Courses, Gender, and Population Division, achieving an accuracy of 31.74%, with 88% precision, and 77% recall. Mason et al. [42] employed academic scores and personal details, yielding comprehensive evaluation metrics: 63% accuracy, 67% precision, 64% recall, 65% F1 score, 26% kappa, and a 64% AUC. Pallathadka et al. [43] examined an ANN-based predictive model using a feature reduction technique, achieving an impressive 92% accuracy rate. In a study involving 682 first-year college students, the Precision Neural Network (PNN) exhibited robust performance with 78% accuracy, 76% responsiveness, and 79% specificity. These results highlight the efficacy of ANN models in forecasting various facets of student performance and their suitability for careers in computer science.

The examination of the cited study highlights areas in which there's room for improvement in career prediction and guidance. Proposed strategies for enhancement revolve around enhancing prediction accuracy by incorporating a broader array of factors into the analysis. These strategies propose expanding the pool of elements to encompass extra demographic, educational, and personal traits, as these aspects play a pivotal role in fine-tuning predictive abilities. This strategy seeks to create more intricate profiles, ultimately refining the accuracy of forecasting suitable career trajectories. Engaging with professionals in the field and undertaking comprehensive research are advised actions for pinpointing pertinent features and refining predictive capacities.

Enhancement of the Skill Set Model, Broadening Prediction Scope, and Overall Career Projection Refinement are necessary. Implementing these enhancements markedly boosts accuracy and expands guidance. Continual research and fine-tuning, driven by feedback, guarantee the model's evolution into a vital instrument for making well-informed decisions about professional paths.

## 3. Materials and methods

### 3.1. Methodology

To fulfill the goals of this suggested investigation, we introduce a research framework illustrated in Fig 1. This framework delineates the progression of methodologies across three primary phases: Feature Fusion and Normalization (FFN), Average Feature Ranking (AFR), and Enhanced Prediction with Weighted Features (PWF). The FFN phase encompasses dataset updates integrating new features, feature extraction (including feature categorization, identification of feature subsets, data cleaning, label encoding, and data scaling), and data splitting.

Additionally, the AFR involves two feature ranking methods: Random Forest, which entails sub-steps such as Bootstrap Table formation, generating multiple Decision Trees, and assigning weights; and Linear Regression, which involves steps like identifying dependent and independent variables, training the algorithm, and retrieving coefficients. Subsequently, the average score of Random Forest's features and coefficients is calculated, followed by feature selection. Enhanced PWF employs Principal Component (PC) for feature reduction, with steps including determining variance ratio and selecting optimal features. Furthermore, Multilayer Perceptron, Activation Functions, and 5×5 fold-cross validation are utilized. These phases lead to an evaluation stage where results are compared with benchmark methods, and performance metrics like Accuracy, Recall, Precision, F1 Score, Error rate, and AUC values are used to assess student performance.

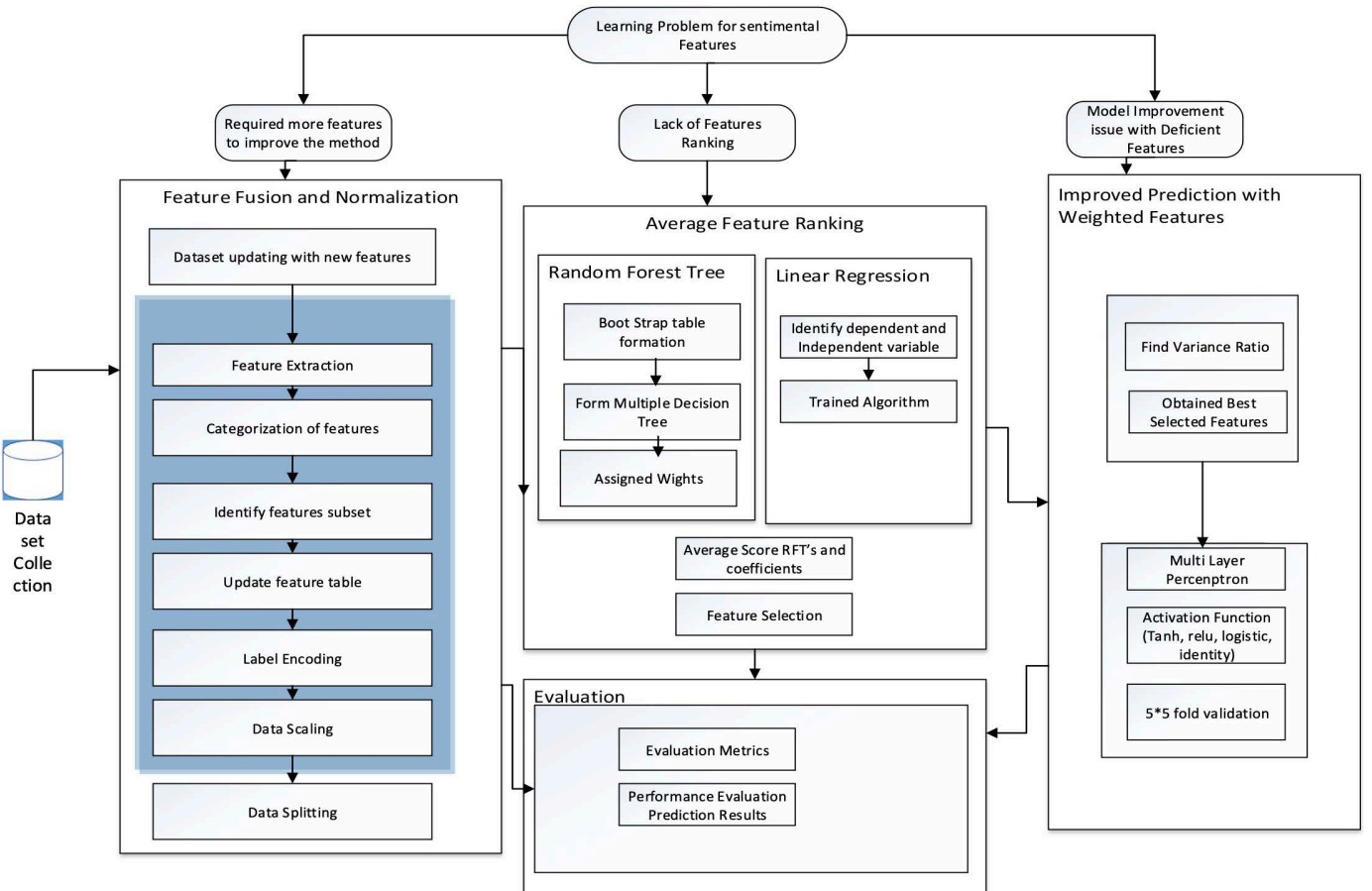

**Fig 1. Framework of Sentiment Assessment System.**

Additionally, the AFR involves two feature ranking methods: Random Forest, which entails sub-steps such as Bootstrap Table formation, generating multiple Decision Trees, and assigning weights; and Linear Regression, which involves steps like identifying dependent and independent variables, training the algorithm, and retrieving coefficients. Subsequently, the average score of Random Forest's features and coefficients is calculated, followed by feature selection. Enhanced PWF employs Principal Component (PC) for feature reduction, with steps including determining variance ratio and selecting optimal features. Furthermore, Multilayer Perceptron, Activation Functions, and 5 × 5 fold-cross validation are utilized. These phases lead to an evaluation stage where results are compared with benchmark methods, and performance metrics like Accuracy, Recall, Precision, F1 Score, Error rate, and AUC values are used to assess student performance.

### 3.2. Feature Fusion and Normalization (FFN)

To enhance career predictions, it's beneficial to gain practical experience and apply skills directly. Consequently, to improve the accuracy of career predictions, additional features are necessary. Consequently, before evaluation, the data underwent preprocessing to address missing values, inconsistencies, and noise. Informative features were used to update the subset of features, and the merging of two input feature vectors into one is termed as feature vector combination. In Fig 2, the FFN process is observed in four different steps. The first step is to update the dataset with new features, in which skill-set features, such as cognitive skills, comfort level, learning styles, and problem-solving skills based on student behavior and characteristics, are added to a data frame. In Step 2, (a) categorization of features, (b) identification of feature subsets, (c) handling missing data, (d) label encoding, and (e) data transformation. During this step, the data were cleaned and converted into numeric form, which caused classification to predict student performance. Next, step 4 split the data into a 30% test set and a 70% training set. In step 5, the normalized dataset was used to evaluate the model's performance. The contribution of the updated feature space is shown in Step 1, as shown in Fig 2. In this study, the skill set model required improvement owing to the need for feature ranking. The feature ranking method requires improvement for identifying the best features and revamping irrelevant features. The top-ranked features are identified with maximum training accuracy. The main objective of ranking the features is to reduce the number of features when performing a prediction. To reduce the number of features required to better understand the data or model. More complex models require considerable computational time for training. Therefore, reducing the number of features requires less time, rendering the model simple and efficient. In brief, four subsets of features are demographic information-based, previous academic record-based, skill-set-based, and behavioral-based input of types to make a new feature vector, and a combination of these methods provides ranked features.

The average Feature Ranking (AFR) procedure is shown in Fig 3, which illustrates the step-by-step process. Step 1 involved input from Phase 1, in which the dataset was updated by adding new features and preprocessing the data, as discussed above. Meanwhile, Step 2 performs the AFR process using Random Forest (RF) and Linear Regression (LR). The RF is the most popular method for checking the rank of each feature. Step 2 involves (a) bootstrap table formation, (b) forming multiple decision trees, and (c) assigning weights to each tree. The results from each tree were aggregated to determine their predictive values. In Step 3, the LR model was fitted to determine the coefficient of each input variable. These coefficients provide the feature importance scores. Step 3 involves the following steps: (a) identifying dependent and independent variables, (b) training algorithm, and (c) retrieving the coefficients.

Step 4 involved assigning feature importance scores from two models to each feature and obtaining the average normalized score. Features with high ranks indicate higher scores, thus reducing the number of features and selecting the best ones. A threshold was set, and features were recursively compared based on their average scores. The feature was remapped to vector feature space if the criterion was met. The top twelve ranked features were identified as optimal for predicting student performance. In Step 5, model performance was assessed using MLP with k-fold cross-validation, which validates effectiveness and trains a subset of input data. K-fold cross-validation divides data into K equal groups,

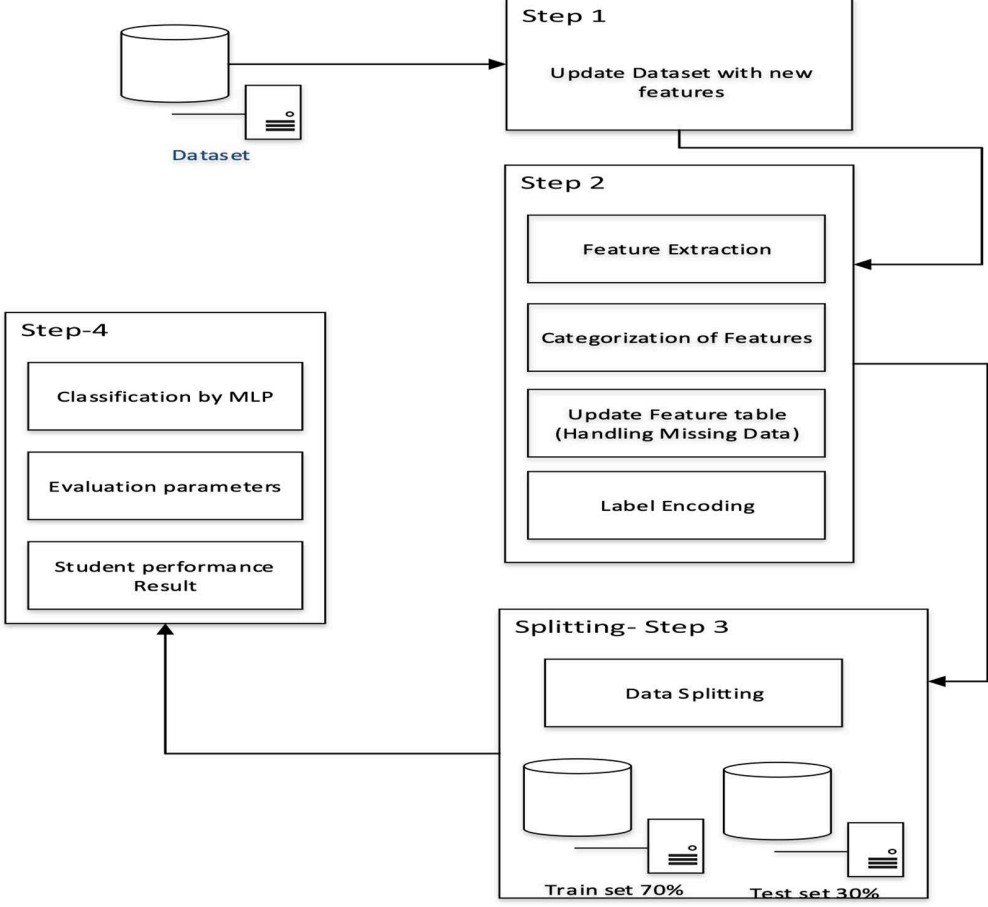

**Fig 2. Normalized Dataset with Skill-Set and Extracted Features.**

using K-1 folds for training and the remaining for testing. This step-by-step feature ranking process improves the model's performance.

### 3.3. Dataset

To conduct experiments, real-time institutional data were gathered from a cited source [9], comprising twenty-four observations from each of the five departments: Humanities, Medical, Engineering, Science, and IT, all from the Career Guidance Dataset of 2022.

$$Accuracy = (TP + TN) / (TP + FP + TN + FN) \tag{1}$$

$$Precision = (TP) / (TP + FP) \tag{2}$$

$$\mathbf{Recall} = (\mathbf{TP})/(\mathbf{TP} + \mathbf{FN}) \tag{3}$$

Therefore, there were a combined total of 574 observations comprising twenty features delineating the information learning patterns of individual students, alongside one target variable denoting students' performance levels. Both datasets were sourced from published materials, and further details regarding the variables can be found in Table 1.

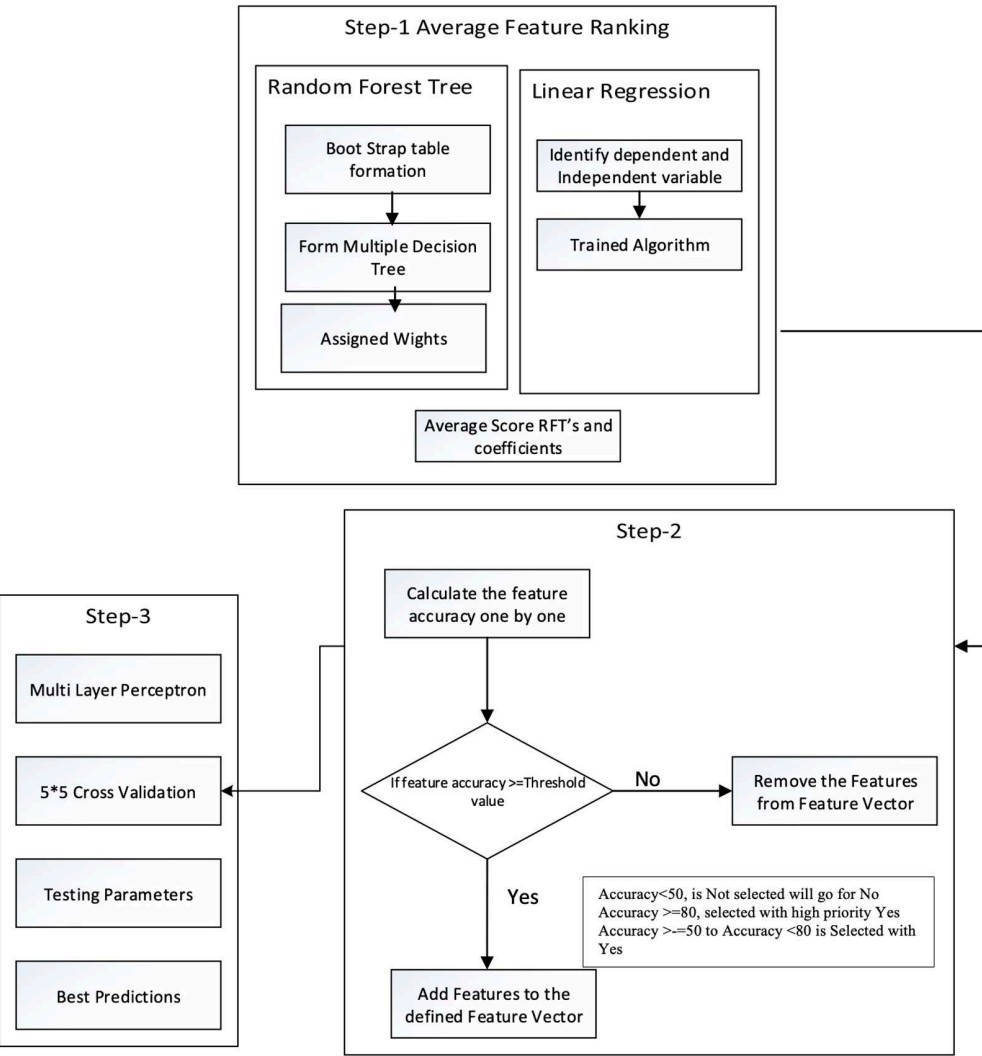

**Fig 3. Average Feature Ranking (AFR) Process.**

## 3.4. Average Feature Ranking (AFR) and Improved Prediction with Weighted Features (PWF)

The prediction range suffers due to compromised attributes, necessitating enhancements in the performance of the selected features. Improved Prediction with Weighted Features (PWF) strives to minimize the feature count for enhancing the classification model. Utilizing three methods—FFN, ARF, and Enhanced PWF—the hybrid combination process leverages their respective strengths. This approach effectively addresses compromised attributes, yielding optimal outcomes in predicting student performance.

Fig 4 indicates the process of Hybrid learning approaches, termed Improved PWF with FFN and ARF. Four steps were involved in predicting student performance in the hybrid method. Step 1 involved preprocessing and updating the data with new features. In Step 2, twelve ranked features out of twenty-four are selected to predict student performance. The next step, Step 3, is to improve the performance of the proposed method, using the feature extraction approach: Principal Component (PC) for feature reduction in this study. In Step 3, the Principal Component (PC) is used to reduce the features and select the best combination of features for improved prediction performance. The PC for dimension reduction

**Table 1. List of Features in Dataset.**

| No. | Features | Types | Description |
|---|---|---|---|
| 1 | Gender | Categorical | Male/Female |
| 2 | Age | Categorical | 18 to 20,21–24, greater than 25 |
| 3 | Family Size | Categorical | Less or equal than 3, more than 3 |
| 4 | Health | Categorical | Healthy, health problem |
| 5 | Marital Status | Categorical | Single, married, other |
| 6 | Work Status | Categorical | Full-time job, part-time job, online job |
| 7 | Guardian | Categorical | Self, father, mother, other |
| 8 | High School Stream | Categorical | Science, Artificial, Art, IT, other |
| 9 | High School Grade | Categorical | 60% to 69%, 70% to 79%, 80% to 89%, greater than 90% |
| 10 | University Type | Categorical | Public/Private |
| 11 | Faculty Type | Categorical | Humanities, Medical, Engineering, Science and IT |
| 12 | Scholarship | Categorical | Yes/No |
| 13 | Transportation Type | Categorical | Bus/Taxi/Private car |
| 14 | Travel time | Categorical | <than 30 mins, 30 mins to 1 hour,> than 1 hour |
| 15 | Study Time (daily) | Categorical | 1 hour, 3 hour, more than 3 hour |
| 16 | Average number of credit hours in one semester | Numerical | 12,15,18,21 |
| 17 | Average Absence days in a course | Categorical | 1 to 2, 3–4, greater than 4 |
| 18 | Failure times (in courses) | Categorical | 0,1,2,3,4 |
| 18 | Internet dependency in Studying | Categorical | To gain more information about study, to gain more understanding of a topic, None |
| 20 | Current Status | Categorical | Graduated/Ungraduated |
| 21 | Cumulative GPA | Categorical | Excellent/Very Good/Good/Acceptable |

generates a new linear combination of features and is used for feature reduction. PC for dimension reduction was used to classify and predict student performance. In the last Step 4, classification was performed by MLP with 5 × 5 fold-cross validation, and the performance was evaluated using different metrics. The third method of Improved PWF steps is shown in Fig 4.

## 4. Results and discussion

This segment evaluates the effectiveness of the suggested AI-driven prediction model employing a combination approach. Statistical outcomes derive from the model's dataset analysis, which encompasses demographic details, academic history, behavioral indicators, and skill sets. The dataset comprises 574 rows and 24 columns and contains features used for classification. These attributes categorize student academic achievement (Cumulative GPA) into Excellent, Good, Very Good, and Acceptable categories.

### 4.1. Evaluation measures

The model generated predictions and evaluated six performance measures: Accuracy, Precision, Recall, F1 score, Error rate, and Area Under the Receiver Operating Characteristic Curve (AUC). It effectively distinguished academically excellent students as such (True Positives - TP) and identified underperforming ones as very good (False Positives - FP). It correctly identified students with unsatisfactory cumulative GPAs as acceptable (True Negatives - TN), but occasionally

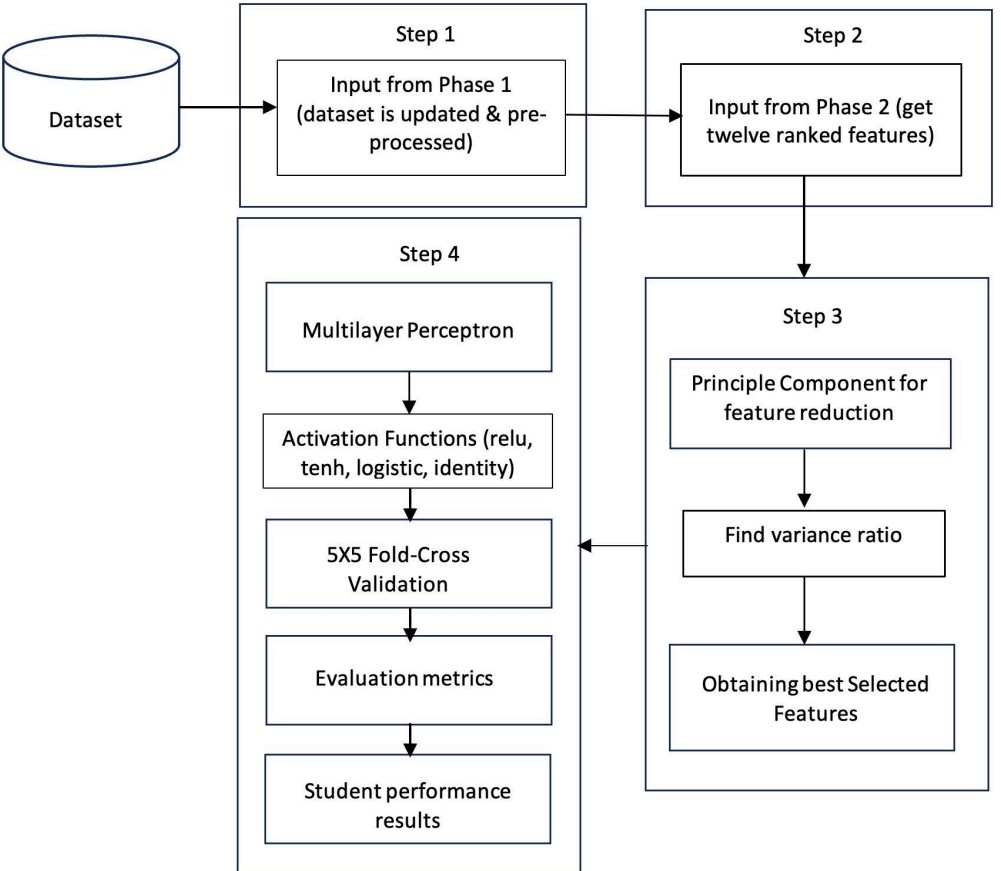

**Fig 4. Hybrid Learning Approaches Process.**

labeled academically strong students as good (False Negatives - FN). Equations (1–3) were utilized to compute Accuracy, Precision, and Recall, respectively.

Accuracy<50, is Not selected will go for No

Accuracy>=80, selected with high priority Yes

Accuracy>-=50 to Accuracy <80 is Selected with Yes

The F1 score is an evaluation parameter that combines precision and recall. Equation (4) shows the formula of the F1 measure to calculate the performance of an algorithm.

$$F1 \; = \; 2\,(Precision \times Recall)\,/\,(Precision \; + Recall)$$
(4)

The F1 measure estimates the classification standard for both excellent and acceptable performance. Equation (5) defines the error rate for the students who are wrongly labeled as acceptable.

$$Error \; = \; (FP \; + \; FN)\,/\,(TP \; + \; TN \; + \; FP \; + \; FN)$$
(5)

The ROC curve measures the number of correct positive classifications achieved with an increase in the rate of false positives. The model's performance was better if the AUC value was larger [17].

## 4.2. Experimental setup

The existing features contain different student behaviors recognized in the formation and updated dataset with new features. The updated dataset contained twenty-four features to predict student performance. Table 2 describes the skill set features.

Table 3 delineates the subsets of features within the student dataset. Those attributes containing personal information about students are categorized under student demographic factors. Previous academic records fall into the realm of student academic information. Given the diverse characteristics and behaviors of students, factors defining their behavior are categorized as behavioral factors. Notably, student skills, closely tied to academic performance, constitute a critical feature set within the student skill-set factors. Data preparation is an integral component of the data analysis process, involving cleaning, transforming, and enhancing data to bolster quality. Due to imperfections in the data collection procedure, the dataset exhibits numerous instances of missing information. To address this, the collected data underwent preprocessing using the Pandas library. Employing a method within Pandas known as data framing mode, the dataset was converted into a series, allowing for the filling of null values with a specified data series.

Following data cleaning, null values were eliminated from the dataset. The algorithm cannot operate on raw data; hence, preprocessing was conducted before applying any classifier. The dataset comprised both categorical and numeric features extracted from Tables 1 and 2. Primarily categorical, these features required conversion into numerical format. The Label Encoder library from Scikit-learn was employed due to the presence of categorical data in both features and labels, facilitating encoding during the preprocessing step. Label Encoder assigns a numerical value ranging from 0 to n-1 to each label. Table 4 illustrates the mapping of target variable values to ordinal data.

When training Linear and ANN models, every input variable undergoes normalization through min-max scaling within the range [0,1], leading to enhanced model performance. These models benefit from the utilization of distance metrics. Additionally, Feature Scaling ensures that larger values do not disproportionately influence the outcome of model fitting. Scikit-learn offers the Min-Max Scalar method to scale each feature within the range of -1–1. Equation (6) demonstrates the process of min-max scaling transformation.

**Table 2. Feature Overview: Types and Descriptions.**

| No. | Features | Types | Description |
|---|---|---|---|
| 1 | Cognitive skills | Categorical | Weak/Strong/Average |
| 2 | Problem-solving skills | Categorical | Excellent/Good/Average |
| 3 | Comfort level | Categorical | Group preferring/Individual preferring |
| 4 | Learning style | Categorical | 1-4, 5-7, 8-10 |

**Table 3. Categories of Features and their Respective Subsets.**

| No. | Categories | Features |
|---|---|---|
| 1 | Demographic Factors | Age, Family size, Marital Status, Health, Work status, and Guardian. |
| 2 | Academic Information | High school stream, High school grade, University type, Faculty type, Study time, Scholarship, Average absence days in a course, Average number of credit hours, Failure times, Current status, and Cumulative GPA (target variable). |
| 3 | Behavioral Factors | Internet dependency in studying, Transportation type, and Travelling time to reach the university. |
| 4 | Skill-set Factors | Cognitive skills, Problem-solving skills, Learning style, and Comfort level. |

**Table 4. Mapping of Cumulative GPA and Target Variable.**

| Current value | New Mapped value |
|---|---|
| Acceptable | 0 |
| Good | 1 |
| Very Good | 2 |
| Excellent | 3 |

$$X'i \;=\; xi - Min(x) \,/\, Max(x) - Max(x) \tag{6}$$

Where min(x) value of the feature is divided by the range of the feature, another form of scaling is called Standardization, where each feature transforms with a mean of 0 and a standard deviation of Equation (7) shows the standardization works as given below.

$$X'i = X - X(mean)/Std(x) \tag{7}$$

Each feature $xx$ is normalized by subtracting the mean and dividing by the standard deviation of the feature set. The FFN process, outlined in Algorithm 1, comprises eight steps. Initially, the dataset is read from the CSV file. Following that, features and target variables are identified within the dataset. The dataset is then augmented with skill-set features. Subsequently, twenty-four features are extracted to delineate categories and feature subsets. Null values in the dataset are checked and filled. Finally, labels are assigned to the target variable, as detailed in Table 4. Adjust the data to a predefined scale, then divide it into training and testing subsets. The MLP classifier underwent training using k-fold cross-validation, yielding results evaluated through metrics such as Accuracy, Recall, Precision, F1 score, Error, and AUC value, presented in Table 5. Additionally, Fig 5 illustrates the comparison between MLP Performance and the FFN method.

### 4.3. Results of Average Feature Ranking (AFR) method

The Random Forest (RF) constructs a forest comprising multiple trees, serving as an ensemble classifier. This method involves growing numerous classification trees, each established using a randomly selected training dataset. The outcomes of these individual trees are combined to assess the predictive value for each observation. RF generates multiple classification trees, employing the bootstrap aggregation (bagging) technique to train each tree from the provided training set.

This approach involves generating multiple decision trees, each with its own specific role. The dataset is divided among these trees, labeled as D1, D2, D3, and so forth, with a designated sample of rows or observations for each. Following training, each decision tree consistently yields accurate results when utilized independently. Upon aggregation, the mean value for each decision tree is computed. The Random Forest tree comprised multiple Decision Trees

**Table 5. Model Performance Metrics After Feature Fusion and Normalization.**

| Run Index | Accuracy | Recall | Precision | F1 Score | Error | AUC |
|---|---|---|---|---|---|---|
| 1 | 0.41 | 0.46 | 0.45 | 0.45 | 0.58 | 0.46 |
| 2 | 0.52 | 0.46 | 0.45 | 0.44 | 0.47 | 0.51 |
| 3 | 0.47 | 0.36 | 0.34 | 0.33 | 0.52 | 0.48 |
| 4 | 0.45 | 0.48 | 0.46 | 0.45 | 0.54 | 0.61 |
| 5 | 0.49 | 0.5 | 0.49 | 0.49 | 0.50 | 0.56 |
| Average | 0.47 | 0.45 | 0.44 | 0.43 | 0.52 | 0.52 |

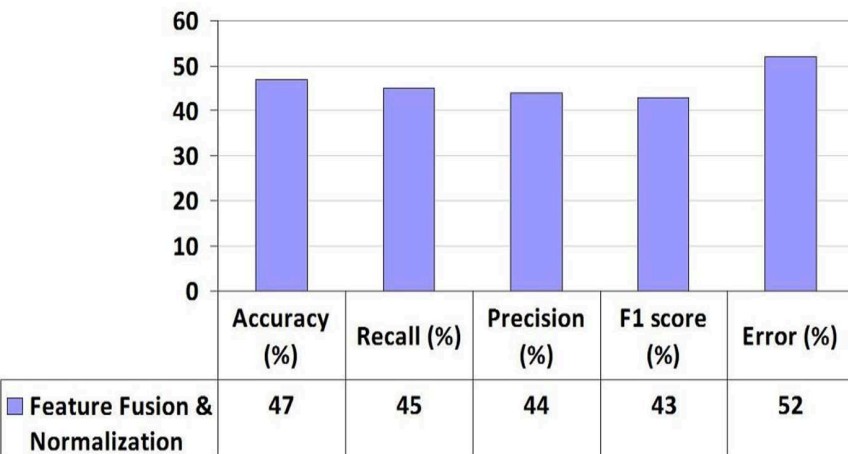

**Fig 5. MLP Performance with Feature Fusion Normalisation Method.**

(DTs), with each DT node evaluated based on the condition of its corresponding feature. Afterward, probabilities were assigned to each node, with each node carrying a specific weight. Higher feature importance correlated with elevated feature values. This classifier was subsequently employed. It uses tree-based strategies that naturally rank the RF and improve node purity. Equations (8–10) show the feature importance calculations of the nodes. Where $f_{ij}$ is the importance of feature $i$, and $n_{ij}$ is the importance of node j. This can be normalized by dividing the sum of all the feature importance values.

**ALGORITHM 1. Method of Average Feature Ranking**

1. *Read Dataset*
2. *Identify Features & Target Label*
3. *Update Dataset with Addition of New Features*
4. *Extract twenty-four Features*
   a. *Identify Category or Types of Features*
   b. *Identify Feature Subsets*
   c. *Check Null Values in Dataset*
   d. *Fill Null Values using fillna()*
   e. *If Input of Student Performance is Acceptable*
   f. *Label = 0*
   g. *Else If Good*
   h. *Label = 1*
   i. *Else If Very Good*
   j. *Label = 2*
   k. *Else Excellent*
   l. *Label = 3*
   m. *Data Scaling of X and y train & test set*
   n. *Xtrain, ytrain, Xtest, ytest = TrainTestSplit(dataset.features, testSize = 0.30)*
5. Train Classifier MLP
   a. *createModel = MLPClassifier (max_iter = 200)*
   b. *y_pred = classifier.predict (X_testscaled)*
   c. *for prediction in Y pred do*
   d. *return scores*
   e. *End*
6. *K-fold Cross Validation*
7. *Evaluate Average Accuracy, Recall, Precision, F1 score, Error, and AUC value*
8. *Results*

$$f_{ij} = \frac{\sum (Node\ J\ Splits\ on\ features\ I)n_{ij}}{\sum K \in all\ node\ In_{ii}}$$

(8)

$$RFfi_i = \frac{normf_i * f_i}{(\sum k) \in j\ all\ features\ I\ f_{ij}}$$

(9)

$$\boldsymbol{f(x)} = \boldsymbol{RF\ f_{ij}} \sum_{n=1}^{j} \boldsymbol{j \in all\ tree\ norm\ I\ f_i/T}$$

(10)

RFfii represents the importance of feature i derived from all trees within the Random Forest (RF) model. norm fij signifies the normalized feature importance for feature i within tree j, and T stands for the total number of trees. Linear machine learning (ML) algorithms ascertain a series of coefficients, wherein the inputs are the weighted sums of predictions. Linear Regression, employed to assess the correlation between dependent and one or more independent variables, utilizes a subset of personal, academic, skill-set, and behavioral features as the independent variables, with cumulative GPA as the dependent variable. Within Linear Regression, the parameters influencing how the model fits the data encompass the number of features encompassed within the dataset and the variety of feature types. While Linear Regression adeptly constructs a model for training, overfitting (exhibiting low bias and high variance) results in suboptimal testing fits. To circumvent overfitting, irrelevant features were pruned, and additional data points were incorporated into the dataset. Multilinear Regression constructs a linear model with coefficients β=(β1, β2,……. Bn) [21].

Linear Regression algorithms fit a model where the prediction is the total weight of input values. Skicit-learn provides a library for determining the feature importance using the Coef method. The Coef method contains the coefficients to predict the target variable by the number of features. Equation (11) calculates the coefficients of each variable as follows:

$$\beta\boldsymbol{i(x)} = \sum_{i=1}^{m} (\boldsymbol{x_m - x'})/(\boldsymbol{y_i - y'})/ \sum_{i=1}^{m} (\boldsymbol{x_i - x'})\boldsymbol{2}$$

(11)

The target variable, denoted as y, is influenced by the independent variable x, with β representing the coefficient of these variables. Feature importance scores were computed for two models, and their normalized averages were determined. Features with higher ranks exhibit scores surpassing others. Equation (12) illustrates the average importance scoring of the features.

$$Avgf_i(i) = \sum_{i=1}^{m} (x_i - x'RFf_i(i))\ /\ \sum_{i=1}^{m} (fi)$$

(12)

Where Avgf_i(i) represents the average function of the feature importance index i, RFfii denotes the factor applied to the importance of the ith value. x_i stands for the ith value of each feature i, and x^'represents the adjusted value of x against the total number of features f. To enhance the interpretability of the proposed model, the feature ranking technique utilized in this study is employed. Table 6 presents the contribution of each feature to the prediction performance. A feature-ranking analysis was conducted to identify the most significant features in the dataset. RF (Random Forest) and LR (Logistic Regression) were utilized for feature selection. Upon fitting the model, RF gauges feature importance via the feature importance property, while LR assesses scores by computing the coefficients of each feature. The outcomes of these methods were aggregated to determine the average score of each feature from the two classification models.

Fig 6 delves into feature importance analysis, comparing three methods: Random Forest (RF), Linear Regression (LR), and the Average Scores approach. This figure provides insights into how these methods evaluate the significance of different features. As demonstrated in Fig 7, based on the average score, the top twelve features, ranked by employing the learning parameter of Average scoring of RF's and Coeff's, include Failure times (courses), Faculty Type, Age, Average number of credit hours in one semester, Cognitive skills, Comfort level, Internet dependency in Studying, Average Absence days in a course, High School Grade, Problem-solving

**Table 6. Feature Importance Scores for Predicting Student Performance Across Models.**

| No. | Features | Random Forest | Linear Regression | Average Scoring |
|-----|----------|---------------|-------------------|-----------------|
| 1 | Gender | 0.0225 | 7.7017 | 0.3218 |
| 2 | Age | 0.0565 | 80.78 | 3.3682 |
| 3 | Family Size | 0.0284 | 3.6098 | 0.1515 |
| 4 | Health | 0.1048 | 7.6846 | 0.3245 |
| 5 | Marital Status | 0.0335 | 2.4633 | 0.104 |
| 6 | Work Status | 0.0312 | 9.3986 | 0.3929 |
| 7 | Guardian | 0.0214 | 10.4348 | 0.4356 |
| 8 | High School Stream | 0.0207 | 6.1734 | 0.258 |
| 9 | High School Grade | 0.0226 | 35.993 | 1.5006 |
| 10 | University Type | 0.0611 | 23.9523 | 1.0005 |
| 11 | Faculty Type | 0.0267 | 33.9405 | 1.4153 |
| 12 | Scholarship | 0.0214 | 15.7876 | 0.6587 |
| 13 | Transportation Type | 0.0537 | 95.712 | 3.9902 |
| 14 | Travel time | 0.0448 | 19.6391 | 0.8201 |
| 15 | Study Time (daily) | 0.0426 | 30.9111 | 1.2897 |
| 16 | Average number of credit hours in one semester | 0.0221 | 83.6771 | 3.4874 |
| 17 | Average Absence days in a course | 0.037 | 66.9216 | 2.7899 |
| 18 | Failure times (in courses) | 0.1449 | 35.1797 | 1.4718 |
| 19 | Internet dependency in Studying | 0.0295 | 94.6277 | 3.944 |
| 20 | Current Status | 0.0402 | 86.565 | 3.6085 |
| 21 | Cognitive skills | 0.0301 | 96.5926 | 4.0259 |
| 22 | Problem-solving skills | 0.488 | 15 | 2 |
| 23 | Comfort level | 0.219 | 50 | 0.6 |
| 24 | Learning style | 0.325 | 92 | 3.8 |

skills, Transportation Type, and Current Status. The findings highlight that student skill sets and academic performance are pivotal in predicting performance, whereas demographic factors wield less influence in forecasting student performance. Algorithm 2 outlines the AFR process. Initially, input was obtained using the FFN method, resulting in twenty-four features for model training. Subsequently, RF and LR were trained to determine the importance of each feature, yielding twelve ranked features surpassing the threshold value. The MLP classifier was then trained utilizing these twelve features and one target variable. Evaluation parameters including Accuracy, Recall, Precision, F1 score, Error, AUC value were computed, and their average values were derived via 5-fold cross-validation, as depicted in Fig 7. It is evident from the results that the model demonstrates enhancement with the incorporation of the top twelve ranked features, yielding an Accuracy of 0.86, Recall of 0.86, Precision of 0.86, F1 Score of 0.86, and Error of 0.13.

### 4.4. Results of improved Prediction with Weighted Feature (PWF) using hybrid methods

The proposed hybrid approach was formulated by amalgamating the aforementioned techniques (Feature Fusion with Ranked Features to enhance prediction Performance with Weighted Features). Table 7 illustrates the performance outcomes of twelve selected features employing the AFR Method. Furthermore, Table 8 presents the confusion matrix of FFN. This table is derived from the accuracy of features processed through the model, with target classes categorized as "acceptable" or "excellent." Here, 0 denotes "acceptable," while 1 signifies "excellent" performance. A single threshold is established for each class: instances with a classification score equal to or exceeding 0.5 are labeled as 1, whereas those scoring below 0.5 are labeled as 0. Regarding the specifics of the table, True Positives (TP) are computed in the first row

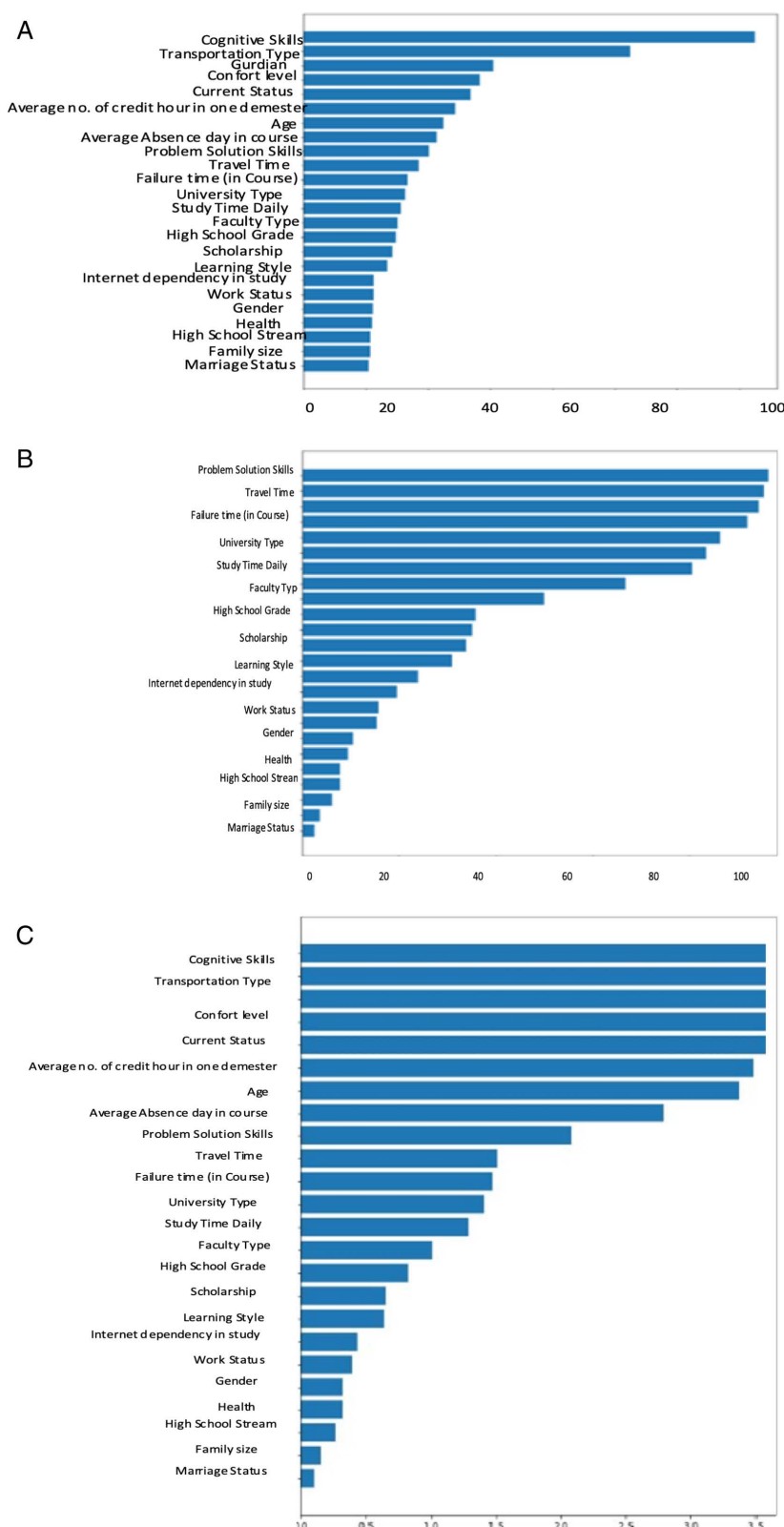

**Fig 6. Comparative Analysis of Feature Importance (a) Features Importance Determined by Random Forest (b) Features Importance Determined by Linear Regression (c) Aggregated Feature Importance (Average Score of Random Forest Trees and Coefficients).**

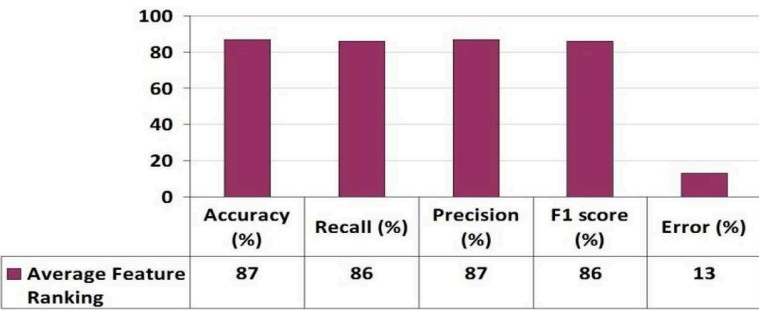

**Fig 7. MLP Performance with Average Feature Rank Method.**

**Table 7. Model Performance Evaluation Summary.**

| Run Index | Accuracy | Recall | Precision | F1 Score | Error | AUC |
|---|---|---|---|---|---|---|
| 1 | 0.86 | 0.85 | 0.85 | 0.85 | 0.13 | 0.96 |
| 2 | 0.87 | 0.89 | 0.89 | 0.89 | 0.12 | 0.93 |
| 3 | 0.86 | 0.85 | 0.85 | 0.85 | 0.13 | 0.89 |
| 4 | 0.84 | 0.87 | 0.88 | 0.87 | 0.15 | 0.93 |
| 5 | 0.89 | 0.85 | 0.85 | 0.85 | 0.10 | 0.91 |
| Average | 0.86 | 0.86 | 0.86 | 0.86 | 0.13 | 0.92 |

and column, True Negatives (TN) are situated to the right of True Positives (TP), False Positives (FP) are positioned in the second column, and False Negatives (FN) are located at the bottom right corner of the matrix. Utilizing these TP, TN, FP, and FN values, various metrics are calculated, encompassing Accuracy, Recall, Precision, and a Score. The formulas for these metrics are provided from Equations 1–4, as referenced earlier.

(a) **Features Importance Determined by Random Forest.**

(b) **Features Importance Determined by Linear Regression.**

(c) **Aggregated Feature Importance (Average Score of Random Forest Trees and Coefficients).**

---

**ALGORITHM 2: Average Feature Ranking (AFR)**

1. *Take Input from FFN Method*
2. *Read 574 × 24 Rows and Columns*
3. *Train Random Forest Classifier*
4. *Calculate rf = feature_importances()*
5. *Train Linear Regression Model*
6. *Calculate lr = model.coefficients*
7. *Take Average Score of RF's and Coefficients*
8. *Avg = (RFfi + βi)/ Total Features*
9. *Select Top Twelve Ranked Features*
10. *Read 574 × 12 Rows and Columns*
11. *Xtrain, ytrain, Xtest, ytest = TrainTestSplit(dataset.features, testSize = 0.30)*
12. *Train Classifier MLP*
13. *createModel (activation function = relu, learning rate = 0.01, max_iter = 200)*
14. *y_pred = classifier.predict(X_test_scaled)*
15. *for y_pred vs y_test do*
16. *Calculate peroramnce metrics*
17. *K-fold Cross Validation*
18. *Evaluate Average Accuracy, Recall, Precision, F1, Error, and AUC value*
19. *Report Results*

---

**Table 8. Comparative Performance Summary of Different Activation Functions.**

| Hyper-parameter Activation function | Run Index | Accuracy | Recall | Precision | F1 Score | Error | AUC |
|---|---|---|---|---|---|---|---|
| relu | 1 | 0.95 | 0.96 | 0.96 | 0.96 | 0.04 | 0.98 |
| relu | 2 | 0.98 | 0.95 | 0.95 | 0.95 | 0.04 | 0.98 |
| relu | 3 | 0.95 | 0.94 | 0.94 | 0.94 | 0.03 | 0.98 |
| relu | 4 | 0.97 | 0.97 | 0.97 | 0.97 | 0.06 | 0.99 |
| relu | 5 | 0.90 | 0.95 | 0.95 | 0.95 | 0.07 | 0.98 |
| relu | Average | 0.95 | 0.95 | 0.96 | 0.95 | 0.05 | 0.98 |
| tanh | 1 | 0.95 | 0.96 | 0.96 | 0.96 | 0.04 | 0.98 |
| tanh | 2 | 0.95 | 0.93 | 0.93 | 0.93 | 0.01 | 0.96 |
| tanh | 3 | 0.96 | 0.93 | 0.93 | 0.93 | 0.04 | 0.98 |
| tanh | 4 | 0.93 | 0.94 | 0.95 | 0.94 | 0.02 | 0.97 |
| tanh | 5 | 0.92 | 0.94 | 0.94 | 0.94 | 0.09 | 0.96 |
| tanh | Average | 0.94 | 0.94 | 0.94 | 0.94 | 0.04 | 0.97 |
| logistic | 1 | 0.93 | 0.91 | 0.95 | 0.91 | 0.06 | 0.94 |
| logistic | 2 | 0.91 | 0.96 | 0.93 | 0.92 | 0.03 | 0.96 |
| logistic | 3 | 0.92 | 0.93 | 0.93 | 0.93 | 0.06 | 0.95 |
| logistic | 4 | 0.95 | 0.94 | 0.92 | 0.94 | 0.04 | 0.96 |
| logistic | 5 | 0.89 | 0.94 | 0.94 | 0.94 | 0.10 | 0.97 |
| logistic | Average | 0.92 | 0.93 | 0.94 | 0.94 | 0.07 | 0.96 |
| identity | 1 | 0.94 | 0.96 | 0.96 | 0.96 | 0.05 | 0.98 |
| identity | 2 | 0.94 | 0.94 | 0.94 | 0.94 | 0.05 | 0.97 |
| identity | 3 | 0.93 | 0.93 | 0.94 | 0.93 | 0.06 | 0.95 |
| identity | 4 | 0.96 | 0.95 | 0.95 | 0.95 | 0.03 | 0.94 |
| identity | 5 | 0.93 | 0.95 | 0.95 | 0.95 | 0.06 | 0.96 |
| identity | Average | 0.94 | 0.95 | 0.95 | 0.95 | 0.05 | 0.97 |

The Improved Prediction with Weighted Feature (PWF) method uses a feature reduction approach, PCA. This prevents overfitting and improves model accuracy. It maps each feature of a given dataset in d-dimensional space to a k-dimensional space, such that $k < d$. PC removed uncorrelated features in a given dataset reduced overfitting problems, and improved classification performance. Each PC coordinates the maximum variance. Eventually, the first component has the maximum variance, and each precedent component has a lower variance value. The PC can be represented in Equation (13) as follows:

$$PC_i = a_1 x_1 + a_2 x_2 + \ldots a_d x_d \tag{13}$$

where PCi is the principal component of i, aj is the numerical coefficient for Xj is the original feature of j parameter of several PC components used for performance prediction. Find the percentage of variance ratio by each of the selected components. It can be normalized by the formula in Equation (14).

$$Explained\_varianc\_ration = \frac{explained\ variance}{\sum explained\ varience} \tag{14}$$

where the sum of explained variance is the original set of features, and explained variances is the amount of variance of each selected component. The set of features is obtained using PC for dimension reduction, and the best set of features is used to predict student performance. The components were selected based on the percentage of variance ratio. PC helps to reduce the dimension of the features and choose the group of the most correlated features. PCA reduced six out of twelve features. The variance ratios of the

top selected six features were 0.05779, 0.05671, 518, 0.05445, 0.05102, 0.04889, and 0.04875. The given values show that Cognitive skills, Comfort level, Internet Dependency in Studying, Failure Time (in courses), and Average Number of Credit Hours in one Semester are the most essential features that improve the given model. The proposed model utilizes the MLP, which is considered the best-suited algorithm for measuring student academic performance for this present study. The classifier used in the MLP is the feed-forward neural network. It has four layers: the input layer, the hidden layer, and the output layer. The MLP is a supervised algorithm with a vital function approximator for prediction and classification problems. There were twenty-four neurons in the input layer, four hidden layers, and one in the output layer, indicating the type of student performance. After loading the dataset, the input and output variables were assigned appropriately. Fine-tuning of the hyperparameters was performed to determine the best sequence of parameters.

The dataset was divided into training and testing datasets by using the train test split utility function. The Grid search is the conventional approach used to tune the optimal parameters of many ML algorithms. The hyperparameters are tuned or optimized to perform the model better. The hyperparameters tune the number of hidden layers, activation function, random state, learning rate, and the maximum number of iterations. The activation functions (relu, tanh, identity, and logistic) and a learning rate of 0.01 with a maximum of 200 iterations were used to train the model. The number of epochs was tuned to train the model and obtain optimal results. The mean Accuracy, Precision, F1 Score, Recall, and ROC values were calculated with the $5 \times 5$ cross-validation method.

Algorithm 3 outlines the process of Enhanced PWF, which comprises nine sequential steps. Initially, input data is acquired through the FFN and AFR methodologies. Subsequently, Principal Component Analysis (PCA) is employed for feature reduction, retaining six features based on their variance ratio. Following this, the dataset is divided into training and testing subsets. The Multi-Layer Perceptron (MLP) classifier is trained using various activation functions, such as relu, tanh, identity, and logistic, utilizing the six selected features alongside a single target variable. Notably, employing the 'relu' activation function yields an Accuracy of 0.95, Recall of 0.95, Precision of 0.96, F1 Score of 0.95, and Error rate of 0.05, as depicted in Fig 8.

The mean values were calculated using 5-fold cross-validation, and the outcomes were derived based on the evaluation metrics including Accuracy, Recall, Precision, F1 score, Error, and AUC value as depicted in Fig 9. It's evident that the 'relu' activation function outperformed the others. The graph in Fig 10 illustrates the performance of three suggested approaches: FFN, AFR, and Hybrid (FFN+AFR+Improved PWF) learning methods. It conspicuously demonstrates that the enhancement across various evaluation metrics led to a notably low error rate, rendering it the most effective predictive model for this prediction task, as indicated in Table 8. Table 9 provides a performance comparison among the proposed FFN, AFR, and Improved PWF methods.

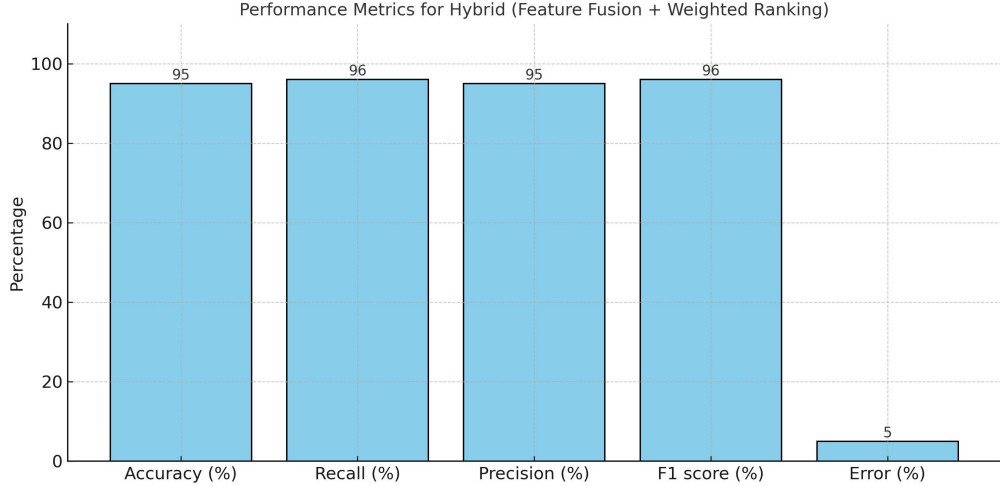

**Fig 8. Performance Evaluation of Tuned Multi-Layer Perceptron with Comparison of Activation Functions.**

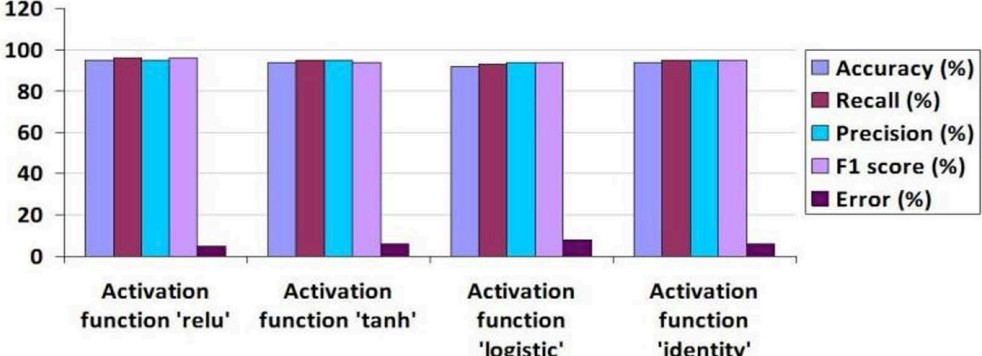

**Fig 9. Results of MLP (ReLU) Performance Using Hybrid Method.**

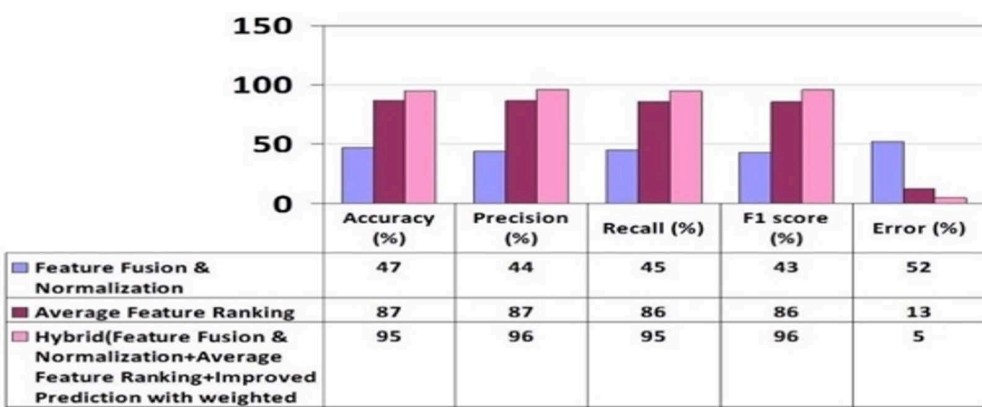

**Fig 10. Overall Performance Results of Evaluation Parameters of Proposed Methods.**

**Table 9. Performance Evaluation of Feature Fusion and normalization, AFR, and Hybrid Methods.**

|  | Accuracy (%) |  | Precision (%) |  | Recall (%) |  | F1 score (%) |  | Error (%) |  |
|---|---|---|---|---|---|---|---|---|---|---|
| Feature Fusion & Normalization | 47 | Low | 44 | Low | 45 | Low | 43 | Low | 52 | High |
| Average Feature Ranking | 87 | Medium | 87 | Medium | 86 | Medium | 86 | Medium | 13 | Low |
| Hybrid (FFN+AFR +Improved PWF) | 95 | High | 96 | High | 95 | High | 96 | High | 5 | Very Low |

## 4.5. Career prediction with accurate ranked features

Predicting one's career trajectory becomes viable by analyzing specific attributes such as skill set and problem-solving prowess. These attributes are identified using the Importance of Features Determined by Random Forest method, and their accuracies are outlined in Table 10. Additionally, factors like Failure time in Courses and faculty study time are extracted using the Importance of Features Determined through weighted Linear Regression, with their respective accuracies also detailed in Table 10. Further features such as Transport type, Comfort level, Current status, cognitive skill value, average study hours, age, and average absences are ranked using the Aggregated Feature Importance method. Their accuracies are provided in Table 11. Fig 11 illustrates the most accurate feature for predicting career fields. This

**Table 10. Career Prediction through Accurately Ranked Features.**

| Propped Technique | Ranked Feature and Profession identification(science/AI/Art/IT) | Ranked Feature Accuracy in percentage |
|---|---|---|
| Importance of Features Determined by Random Forest | Cognitive skill | 97 |
| Importance of Features Determined by Random Forest | Transport type | 80 |
| Importance of Features Determined through weighted Linear Regression | Problem Solving Skills | 100 |
| Importance of Features Determined through weighted Linear Regression | Travel Time | 99 |
| Importance of Features Determined through weighted Linear Regression | Failure time in course | 98 |
| Importance of Features Determined through weighted Linear Regression | University/Faculty Type | 97 |
| Importance of Features Determined through weighted Linear Regression | Study Time daily | 94 |
| Aggregated Feature Importance (Average Score of Random Forest Trees and Coefficients) | Cognitive skill | 100 |
| Aggregated Feature Importance (Average Score of Random Forest Trees and Coefficients) | Transport type | 100 |
| Aggregated Feature Importance (Average Score of Random Forest Trees and Coefficients) | Confort level | 100 |
| Aggregated Feature Importance (Average Score of Random Forest Trees and Coefficients) | Current Status | 100 |
| Aggregated Feature Importance (Average Score of Random Forest Trees and Coefficients) | Averagre number of credit hours in current semester | 100 |
| Aggregated Feature Importance (Average Score of Random Forest Trees and Coefficients) | Age | 95 |
| Aggregated Feature Importance (Average Score of Random Forest Trees and Coefficients) | Average absents in a day | 97 |

**Table 11. Evaluation of Higher Rank Features using MLP.**

| Technique | Feature Name | Precision | Recall | Average |
|---|---|---|---|---|
| Average Score of Random Forest Trees and Coefficients | Skill Set | 97 | 97 | 0.970874 |
| Importance of Features Determined through weighted Linear Regression | Problem solving | 97 | 97 | 0.970874 |
| Importance of Features Determined through weighted Linear Regression | Travel Time | 94 | 0.9428 | 0.955953 |
| Importance of Features Determined through weighted Linear Regression | Failure in Course | 94 | 0.943857 | 0.955653 |

assessment aids in determining the suitability of various professions such as art, engineering, IT, science, AI, etc., by evaluating their applicability based on cognitive skill as the root node and conducting comparisons. The efficacy of MLP evaluation with the top four features, following the implementation of three enhancement techniques, is presented for Career prediction in Table 11.

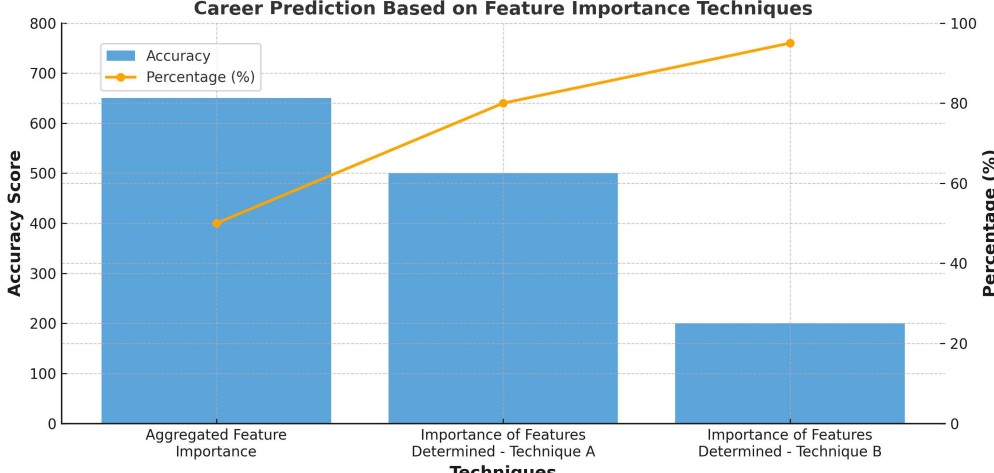

**Fig 11. Career Prediction through accurate features.**

---

**ALGORITHM 3: Improved Prediction with Weighted Features**

1. Take Input From FFN & AFR Methods
2. Read 574 × 12 Rows and Columns
3. Use Principal Component for Feature Reduction
4. Six Features Selected with their variance ratio
5. Read 574 × 6 Rows and Columns
6. Xtrain, ytrain, Xtest, ytest = TrainTestSplit(dataset.features, testSize = 0.30)
7. Train Classifier MLP
   A. createModel (activation function = relu/tanh/identity/logistic, learning rate = 0.01, max_iter = 200)
   B. y_pred = classifier.predict(X_testscaled)
   C. for prediction in Y pred do
   D. return scores E. End
8. K-fold Cross Validation
9. Improved Comparative Parameters Values Achieved
   A. Tuned Parameters with different Activation Function Evaluate Average Accuracy Value
   B. Tuned Parameters with different Activation Function Evaluate the Average Recall Value
   C. Tuned Parameters with different Activation Function Evaluate Average Precision Value
   D. Tuned Parameter with different Activation Functions Evaluate Average F1 score Value
   E. Tuned Parameter with different Activation Functions Evaluate Average Error rate Value
   F. Tuned Parameters with different Activation Function Evaluate Average AUC Value
10. Results

---

## 5. Comparative analysis

Tables 10, 12, 13 present a comparative analysis of evaluation metrics between our proposed method and benchmark studies. Table 14 indicates a 5% increase in Accuracy and a 6% decrease in error rate compared to the FFN method. Moving to Table 12, we note an 8% enhancement in Accuracy, a 9% improvement in Recall, a 5% rise in Precision, a 7% increase in F1 Score, and an 8% reduction in error rate relative to the AFR method. Additionally, Table 13 demonstrates significant improvements with an 11% increase in Accuracy, a 2% rise in Recall, a 9% improvement in Precision, an impressive 42% boost in F1 Score, and a 10% reduction in error rate with the hybrid method.

Figs 12–14 compare our proposed method with benchmark studies across evaluation parameters, including Accuracy, Recall, Precision, F1 Score, and Error Rate. All three approaches exhibit enhancements over previous studies, with our proposed method showing the best performance and the lowest error rate among them.

**Table 12. Comparison of Evaluation Parameters with Benchmark Study-2.**

| Author | Features | Algo-rithm | Accu-racy (%) | Recall (%) | Preci-sion (%) | F1 Score (%) | Error (%) |
|---|---|---|---|---|---|---|---|
| Mengash, Hanan et al. 2020 | High school average grade, scholastic achievement in admission test score, and aptitude test score. | MLP | 79 | 78 | 81 | 79 | 21 |
| Proposed study | Demographic factors, academic information, behavioral, and skills-set features | MLP | 87 | 87 | 86 | 86 | 13 |

**Table 13. Comparison of Evaluation Parameters with Benchmark Study-3.**

| Author | Features | Algo-rithm | Accu-racy (%) | Recall (%) | Preci-sion (%) | F1 Score (%) | Error (%) |
|---|---|---|---|---|---|---|---|
| Sun, E T et al. 2019 | Demographics, high school scores and combined scores of examinations | MLP | 84 | 94 | 86 | 54 | 15 |
| Proposed study | Demographic factors, academic informa-tion, behavioral, and skills-set features | MLP | 95 | 96 | 95 | 96 | 5 |

**Table 14. Comparison of Evaluation Parameters with Benchmark Study-1.**

| Author | Features | Algo-rithm | Accu-racy (%) | Recall (%) | Preci-sion (%) | F1 Score (%) | Error (%) |
|---|---|---|---|---|---|---|---|
| Alsalman, Yasmeen Shaher et al. 2019 | Student academic record, background information, and behavior features | MLP | 42 | NaN | NaN | NaN | 58 |
| Proposed study | Demographic factors, academic informa-tion, behavioral, and skills-set features | MLP | 47 | 45 | 44 | 43 | 52 |

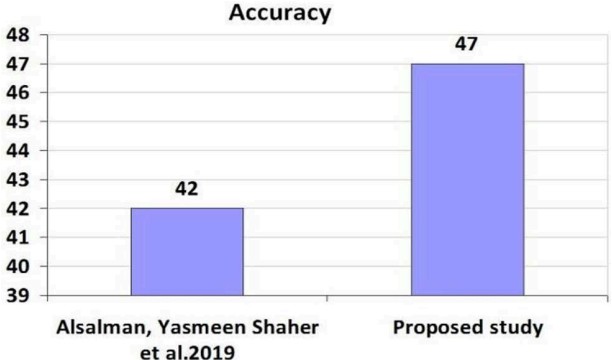

**Fig 12. Comparison Results of Accuracy Values with First Benchmark.**

## 5.1. Comparison with X boost technique

The proposed approach is compared with a reference study [44], where the reference study achieved an accuracy of 77% using XGBoost for student career prediction. In contrast, the proposed hybrid technique significantly improves accuracy to 95%. As shown in Fig 15, the comparison highlights the superiority of the proposed approach over the state-of-the-art method. The proposed model is particularly effective for capturing non-linear data patterns, whereas XGBoost struggles due to its limited ability to model complex feature interactions. The primary reason for XGBoost's lower performance

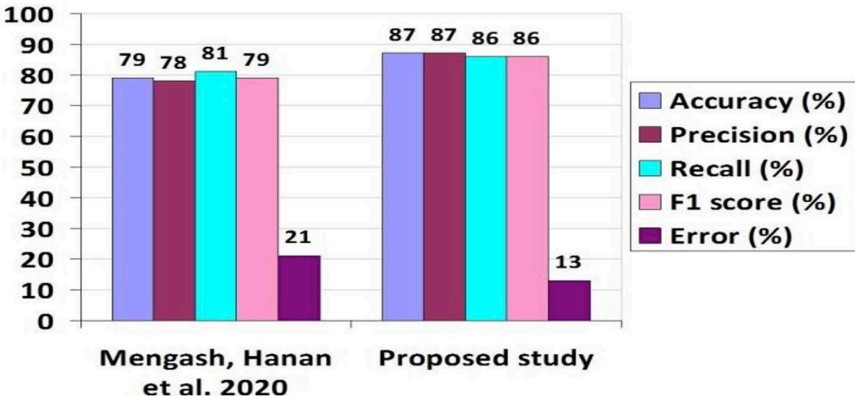

**Fig 13. Comparison Results of Evaluation Parameters Values with the second Benchmark.**

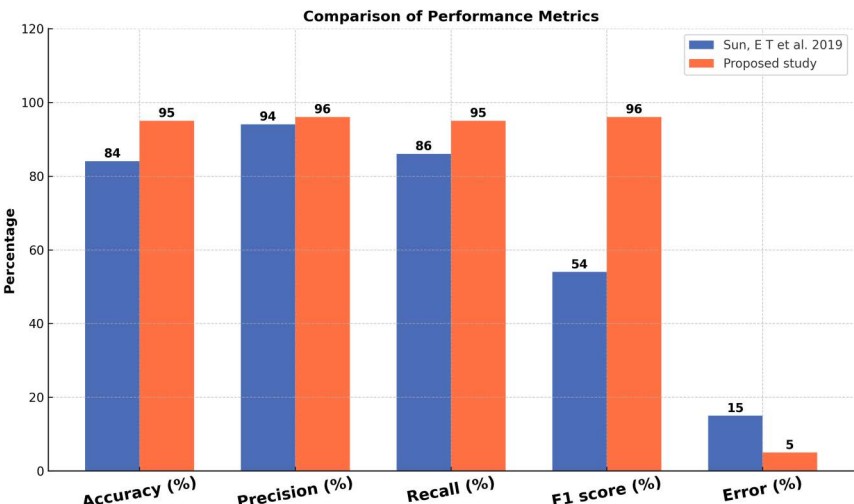

**Fig 14. Comparison Results of Evaluation Parameters Values with Third Benchmark.**

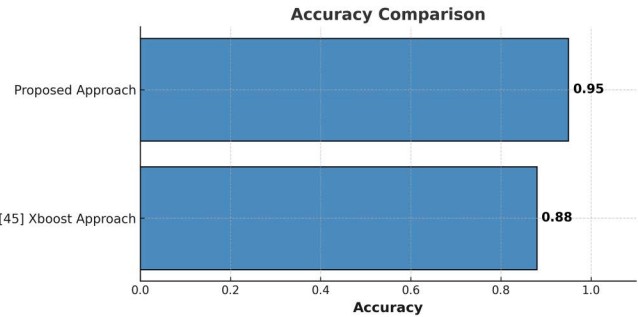

**Fig 15. Career Prediction Comparison with X Boost approach and the Proposed Approach.**

compared to the proposed approach lies in its inability to deeply analyze feature patterns essential for accurate student career prediction.

## 6. Conclusion and future work

This study introduced a comprehensive strategy for data preprocessing and feature selection that notably enhanced the performance of the predictive model. The Feature Fusion and Normalization (FFN) technique enriched the dataset by integrating skill-set features and applying subsequent preprocessing steps to improve data quality. The Average Feature Ranking (AFR) method effectively pinpointed the most important features, resulting in a highly accurate prediction model. Additionally, the Improved Prediction with Weighted Feature (PWF) method, which utilized Principal Component Analysis and advanced machine learning techniques, further refined the model's predictive abilities. The combined use of FFN, AFR, and Improved PWF led to superior prediction performance, achieving an Average Accuracy of 95%, F1 Score of 96%, Recall of 96%, Precision of 95%, and an impressive AUC value of 98%. These results significantly surpassed those of the benchmark study, demonstrating the effectiveness of the proposed methodology in enhancing predictive accuracy and model reliability.

The proposed study centers on a restricted range of skill-set characteristics. It's advised to integrate supplementary attributes linked to skill-set features to enrich the model's comprehension. Assessment of the model in this investigation encompassed metrics such as Accuracy, Precision, Recall, and F1 score. Two methods for assessing feature importance were employed. Subsequent research could delve into integrating innovative feature importance methodologies, particularly harnessing deep learning neural networks. Comparing outcomes derived from these fresh techniques could yield valuable insights for future studies.

## Author contributions

**Conceptualization:** Sana Yaqoob, Md Imran Alam, khurram Ejaz.

**Data curation:** Sana Yaqoob, Mohammad Zubair Khan, Anmol Ejaz, khurram Ejaz.

**Formal analysis:** khurram Ejaz.

**Funding acquisition:** Ayman Noor, Talal H. Noor, Anmol Ejaz.

**Investigation:** Nadim Rana.

**Methodology:** Nadim Rana.

**Project administration:** khurram Ejaz.

**Resources:** Ayman Noor, Talal H. Noor, Anmol Ejaz, Md Imran Alam.

**Software:** Md Imran Alam.

**Supervision:** khurram Ejaz.

**Validation:** Nadim Rana.

**Visualization:** Ayman Noor, Nadim Rana.

**Writing – original draft:** Sana Yaqoob.

**Writing – review & editing:** Sana Yaqoob, Anmol Ejaz, Nadim Rana, khurram Ejaz.

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
