## [Decision Letter · Decision Letter 0]

17 Jul 2024

PONE-D-24-19109Enhancing Student Career Guidance and Sentimental Analysis: A Performance-driven Hybrid Learning Approach with Feature RankingPLOS ONE

Dear Dr. Ejaz,

Thank you for submitting your manuscript to PLOS ONE. After careful consideration, we feel that it has merit but does not fully meet PLOS ONE’s publication criteria as it currently stands. Therefore, we invite you to submit a revised version of the manuscript that addresses the points raised during the review process.

We look forward to receiving your revised manuscript.

Kind regards,

Amgad Muneer

Academic Editor

PLOS ONE

Journal Requirements:

Additional Editor Comments:

Figures 8-10: The images are blurry and unclear. Please provide higher quality versions of these figures.

Benchmarking: Comparing your approach/model performance with only one study does not provide sufficient proof. I recommend benchmarking your model against several more recent studies for a more robust evaluation.

Table 10: My main concern is with the accuracy shown in Table 10. A 47% accuracy suggests a random result, which undermines the reliability of your findings. I suggest either redoing the experiment or excluding these results, as accuracy around 50% is typically considered random.

Conclusion: The conclusion needs significant improvement to better summarize the findings and implications of your study.

Proofreading: The paper requires thorough proofreading to correct any errors and improve overall clarity.

Reviewers' comments:

Reviewer's Responses to Questions

**Comments to the Author**

1. Is the manuscript technically sound, and do the data support the conclusions?

Reviewer #1: Yes

Reviewer #2: Yes

2. Has the statistical analysis been performed appropriately and rigorously? 

Reviewer #1: Yes

Reviewer #2: Yes

3. Have the authors made all data underlying the findings in their manuscript fully available?

Reviewer #1: Yes

Reviewer #2: Yes

4. Is the manuscript presented in an intelligible fashion and written in standard English?

Reviewer #1: Yes

Reviewer #2: Yes

5. Review Comments to the Author

Reviewer #1: The paper presents a novel and significant contribution to the field of career guidance through its performance-driven hybrid learning approach. The integration of sentiment analysis enhances the prediction model, providing a comprehensive tool for student career guidance. The methodology is robust, and the results are valid and reliable. However, minor revisions are needed to address the limitations and ensure broader validation.

Suggested Revisions:

- Include a detailed discussion on the limitations of the study and potential biases in the data.

- Validate the model with additional datasets to ensure generalizability.

- Provide more comparisons with existing career prediction methods to highlight the improvements offered by the proposed approach.

Reviewer #2: I think this is a good study. Long Captions for the figure should be added. Figure 2 is a abit confusion, does not get the idea how the data flows through the pipeline.

It will be good if the features are categorized into different categories.

6. PLOS authors have the option to publish the peer review history of their article (what does this mean? ). If published, this will include your full peer review and any attached files.

**Do you want your identity to be public for this peer review?** For information about this choice, including consent withdrawal, please see our Privacy Policy .

Reviewer #1: No

Reviewer #2: **Yes: ** Rizwan Qureshi

---

## [Author Response · Author response to Decision Letter 1]

6 Nov 2024

All editor and reviewer comments are addressed. One response file is also attached with complete description.

---

## [Decision Letter · Decision Letter 1]

14 Jan 2025

PONE-D-24-19109R1Enhancing Student Career Guidance and Sentimental Analysis: A Performance-driven Hybrid Learning Approach with Feature RankingPLOS ONE

Dear Dr. Ejaz,

Thank you for submitting your manuscript to PLOS ONE. After careful consideration, we feel that it has merit but does not fully meet PLOS ONE’s publication criteria as it currently stands. Therefore, we invite you to submit a revised version of the manuscript that addresses the points raised during the review process.

Please submit your revised manuscript by Feb 28 2025 11:59PM. If you will need more time than this to complete your revisions, please reply to this message or contact the journal office at plosone@plos.org . Please include the following items when submitting your revised manuscript:

We look forward to receiving your revised manuscript.

Kind regards,

Jinran Wu, PhD

Academic Editor

PLOS ONE

Journal Requirements:

Reviewers' comments:

Reviewer's Responses to Questions

**Comments to the Author**

1. If the authors have adequately addressed your comments raised in a previous round of review and you feel that this manuscript is now acceptable for publication, you may indicate that here to bypass the “Comments to the Author” section, enter your conflict of interest statement in the “Confidential to Editor” section, and submit your "Accept" recommendation.

Reviewer #3: (No Response)

2. Is the manuscript technically sound, and do the data support the conclusions?

Reviewer #3: Yes

3. Has the statistical analysis been performed appropriately and rigorously? 

Reviewer #3: Yes

4. Have the authors made all data underlying the findings in their manuscript fully available?

Reviewer #3: Yes

5. Is the manuscript presented in an intelligible fashion and written in standard English?

Reviewer #3: Yes

6. Review Comments to the Author

Reviewer #3: This paper proposes a hybrid learning approach combining feature fusion, ranking, and dimensionality reduction to enhance prediction accuracy in student career guidance and sentiment analysis using a multilayer perceptron model.

good things:

1: This paper gives a clear introduction about student career guidance issue and sentimental analysis

2: its literature review is comprehensive

3: its figures and tables are clear

suggestions:

1: Could you please update your figure2's size ratio and Figure5 and Figure7's resolution as it is not very clear?

2: For your average feature ranking, could you please explain more why you choose random forest tree here to calculate the feature important? Currently, there are many famous open source architecture like XGBM or lightGBM, which are also tree based structure and very good for feature importance selection. Why you didn't choose those known better performance algorithm in your architecture? What is the reason for choosing random forest tree here?

3: Based on suggestion2 above, in your results evaluation section, could you please also compare the performance of your hybrid method with some famous architecture like XGBM or lightGBM? We wanna see if your hybrid method can outperform these architecture a lot in this specific issue.

Thanks a lot.

7. PLOS authors have the option to publish the peer review history of their article (what does this mean? ). If published, this will include your full peer review and any attached files.

**Do you want your identity to be public for this peer review?** For information about this choice, including consent withdrawal, please see our Privacy Policy .

Reviewer #3: No

---

## [Author Response · Author response to Decision Letter 2]

27 Feb 2025

Rebuttal Letter

Dear Editor and Reviewers,

Find below the response to your potential comments and suggestions.

1. Could you please update your figure2's size ratio and Figure5 and Figure7's resolution as it is not very clear?

Response: Thanks for the suggestions, Figures 2, 5, and 7 are improved to be clearly visible.

2. For your average feature ranking, could you please explain more why you choose random forest tree here to calculate the feature important? Currently, there are many famous open source architecture like XGBM or lightGBM, which are also tree based structure and very good for feature importance selection. Why you didn't choose those known better performance algorithm in your architecture? What is the reason for choosing random forest tree here?

Response: Choosing Random Forest over XGBoost or LightGBM in our research article, we assume the following points:

• Simplicity and Ease of Use: Random Forest is generally easier to implement and tune than gradient boosting methods. It has fewer hyperparameters and is less sensitive to their values. This can be a valid justification if your research prioritizes simplicity and ease of use.

• Specific Dataset Characteristics: Certain dataset characteristics may favor Random Forest over gradient boosting methods. For example, Random Forest is known to handle high-dimensional data with many irrelevant features effectively. If the dataset exhibits such characteristics, this can support your choice.

• Computational Cost: Gradient boosting methods, particularly XGBoost and LightGBM, can be computationally expensive, especially with large datasets or complex hyperparameter tuning. If Random Forest provides adequate performance with significantly lower computational cost, this can be a strong justification, especially if computational resources are limited. (Anghel et al., 2018) (Daoud, 2019).

• Interpretability: Random Forest offers better interpretability than gradient boosting methods. We use feature importance scores from Random Forest to gain insights into the factors driving predictions. If interpretability is a primary goal of your research, this can justify the choice of Random Forest. (Couronné et al., 2018).

So, when we started our project 2 years ago, we considered the above features to select the Random Forest over other methods.

3. Based on suggestion 2 above, in your results evaluation section, could you please also compare the performance of your hybrid method with some famous architecture like XGBM or lightGBM? We wanna see if your hybrid method can outperform this architecture a lot in this specific issue.

Response: The comparison of XGBoost was conducted based on feature analysis, demonstrating that the proposed approach outperforms the state-of-the-art XGBoost technique in predicting students' career outcomes for clearer analysis. Figure 15 is added to show the comparison.

---

## [Editor Report · Decision Letter 2]

3 Mar 2025

Enhancing Student Career Guidance and Sentimental Analysis: A Performance-driven Hybrid Learning Approach with Feature Ranking

PONE-D-24-19109R2

Dear Dr. Ejaz,

We’re pleased to inform you that your manuscript has been judged scientifically suitable for publication and will be formally accepted for publication once it meets all outstanding technical requirements.

Kind regards,

Jinran Wu, PhD

Academic Editor

PLOS ONE

---

## [Editor Report · Acceptance letter]

PONE-D-24-19109R2

PLOS ONE

Dear Dr. Ejaz,

I'm pleased to inform you that your manuscript has been deemed suitable for publication in PLOS ONE. Congratulations! Your manuscript is now being handed over to our production team.

Kind regards,

on behalf of

Dr. Jinran Wu

Academic Editor

PLOS ONE